# New Diterpenes and Diterpene Glycosides with Antibacterial Activity from Soft Coral *Lemnalia bournei*

**DOI:** 10.3390/md22040157

**Published:** 2024-03-29

**Authors:** Xiao Han, Huiting Wang, Bing Li, Xiaoyi Chen, Te Li, Xia Yan, Han Ouyang, Wenhan Lin, Shan He

**Affiliations:** 1Li Dak Sum Yip Yio Chin Kenneth Li Marine Biopharmaceutical Research Center, Ningbo University, Ningbo 315211, China; hanxxlee@163.com (X.H.); ting2531680215@126.com (H.W.); 18868105112@163.com (B.L.); chennimo@163.com (X.C.); telinbu@163.com (T.L.); yanxia@nbu.edu.cn (X.Y.); 2Department of Marine Pharmacy, College of Food Science and Engineering, Ningbo University, Ningbo 315800, China; 3Institute of Drug Discovery Technology, Ningbo University, Ningbo 315211, China; 4Ningbo Institute of Marine Medicine, Peking University, Ningbo 315800, China; whlin@bjmu.edu.cn

**Keywords:** soft coral, *Lemnalia bournei*, diterpenes, diterpene glycosides, antibacterial activity

## Abstract

Five new biflorane-type diterpenoids, biofloranates E–I (**1**–**5**), and two new bicyclic diterpene glycosides, lemnaboursides H–I (**6**–**7**), along with the known lemnabourside, were isolated from the South China Sea soft coral *Lemnalia bournei*. Their chemical structures and stereochemistry were determined based on extensive spectroscopic methods, including time-dependent density functional theory (TDDFT) ECD calculations, as well as a comparison of them with the reported values. The antibacterial activities of the isolated compounds were evaluated against five pathogenic bacteria, and all of these diterpenes and diterpene glycosides showed antibacterial activities against *Staphylococcus aureus* and *Bacillus subtilis*, with MICs ranging from 4 to 64 µg/mL. In addition, these compounds did not exhibit noticeable cytotoxicities on A549, Hela, and HepG2 cancer cell lines, at 20 μM.

## 1. Introduction

Marine organisms constitute a treasure trove of biologically active natural products. Soft corals are particularly noteworthy for their rich array of secondary metabolites. The genus *Lemnalia* within soft corals (Coelenterata, Octocorallia, Alcyonacea) is renowned for its diverse terpenoid compounds [1,2,3,4,5], including sesquiterpenes, diterpenoids, diterpene glycosides, and steroids, which exhibit significant biological activities. Sesquiterpenoids, in particular, emerged as the predominant and characteristic metabolites of the *Lemnalia* genus due to their plentiful occurrence and structural diversity [1,6,7,8]. In contrast, biflorane-based diterpenoids are less prevalent, forming minor constituents in *Lemnalia* soft coral [2,5].

Significantly, the secondary metabolites derived from the *Lemnalia* genus frequently feature innovative carbon atom linkages and substituent configurations. Prior research indicates that *Lemnalia*-derived diterpenoids exhibit a variety of ring structures, including biflorane, 5,3,6-tricyclic, 5,7,3-tricyclic, and 10-membered carbocyclic frameworks, which are biogenetically interconnected [2,5]. Geranylgeranyl pyrophosphate (GGPP, C20) is considered the common biosynthetic precursor for these diterpenoids. Specifically, lemnaboursides from *Lemnalia bournei* are diterpene glycosides, featuring D-glucose linked to a diterpene aldehyde via an acetal bond, acetylated or modified by the lemnal-1(10)-ene-7,12-diol group on the sugar moiety, or as derivatives with opened ring D [9,10,11,12]. Diterpenoids and diterpene glycosides demonstrate a wide range of biological activities, including antibacterial, antiviral, and anti-inflammatory effects [2].

As part of our ongoing program to search for biologically active compounds from marine organisms, we collected the soft coral *Lemnalia bournei* off the coast of Xisha Island in April 2021. Chemical investigation of the acetone extract led to the isolation and identification of five new biflorane-type diterpenoids, named biofloranates E–H (**1**–**5**), and two new bicyclic diterpene glycosides, named lemnaboursides H–I (**6**–**7**) (Figure 1). This paper describes the isolation, structural elucidation, and antibacterial activity evaluation of these isolates.

## 2. Results

Biofloranate E (**1**) was isolated as a colorless oil. Its molecular formula was determined to be C_20_H_36_O_2_ using HRESIMS (*m*/*z* 291.2669 [M − H_2_O + H]^+^ (calcd for C_20_H_35_O_2_, 291.2682), requiring three degrees of unsaturation. The ^1^H and ^13^CNMR spectra (Table 1) showed 20 resonances attributable to two sp^2^ olefinic carbons (one CH and one quaternary) and eighteen sp^3^ carbons (four CH_3_, eight CH_2_, five CH, and one C), including two oxygenated ones, accounting for one of the three degrees of unsaturation suggested by the molecular formula (Appendix A). The remaining two degrees of unsaturation indicated that compound **1** had to be bicyclic. The bicyclic biflorane framework was established through comprehensive 2D NMR analysis [5]. The ^1^H-^1^H COSY spectrum delineated the proton connectivities between H_2_-2/H_2_-1/H-10/H-5/H-6/H_2_-7/H_2_-8, H-6/H-11/H_2_-12/H_2_-13/H_2_-14/H-15/H_2_-16, H-11/H_3_-18, and H-15/H_3_-17. These data together with the key HMBC correlations (Figure 2), such as H_2_-1, H_2_-2, H_3_-19/C-3 and H_2_-1, H_2_-7, H_2_-8, H_3_-20/C-9, facilitated the elucidation of the planar structure of **1**.

The relative configuration of **1** was ultimately determined using 1D NOE spectroscopy. NOEs observed between H-5 and H_3_-18, and H-10 and H-5, indicated that H-5, H-10, and the 11-Me group are on the same side of the molecule (Figure 3). The lack of NOE interactions between H-5/H-11, H-6/H_3_-18, and H-10/H_3_-20 suggested that H-6, H-11, and the 9-Me group are oriented on the opposite face. The absolute configurations of **1** were deduced using TDDFT/ECD calculations. The Boltzmann-averaged ECD spectrum for the (5*R*, 6*S*, 9*S*, 10*S*, 11*R*) configuration of **1** closely matched the experimental spectrum, as anticipated (Figure 4a). The ECD spectroscopic data, in conjunction with the biogenetic analysis, confirmed the absolute configuration of compound **1** as 5*R*, 6*S*, 9*S*, 10*S*, 11*R*. However, the configuration at C-15 remained elusive.

Biofloranate F (**2**) was also obtained as a colorless oil. The HRESIMS analysis indicated a molecular formula of C_20_H_32_O_2_ for **2**, with an ion peak at *m*/*z* 287.2371 [M − H_2_O + H]^+^ (calcd for C_20_H_31_O, 287.2375), suggesting five degrees of unsaturation. The molecular formula of 2, when compared to compound **1**, indicated a deficiency of four hydrogens, which is consistent with an additional double bond (*δ*_C_/*δ*_H_ 155.2/6.46 and *δ*_C_139.3) and an aldehyde group (*δ*_C_/*δ*_H_ 195.5/9.37). The ^1^H-^1^H COSY cross-peak between H_2_-13/H-14, along with the HMBC correlation of H-14/C-12/C-13/C-15/C-16/C-17 (Figure 2), confirmed the presence of an additional double bond between C-14 and C-15 in **2**. The location of the aldehyde group was determined through the HMBC correlations from H-16 to C-14, C-15 and C-17, from H_3_-17 to C-16, and from H-14 to C-16. Consequently, the planar structure of **2** was established. The distinct NOE correlations between H-14 and H-16 suggested the *E* geometry for the trisubstituted double bond Δ14(15). The relative stereochemistry of **2** was also assigned by 1D NOE spectra analysis between H_3_-18/H-5, H_3_-20/H-5, and H_3_-20/H-10, and no NOE could be detected between H_3_-18 and H-6. Based on these findings, the relative configuration of **2** was determined to be 5*R**, 6*S**, 9*R**, 10*S**, 11*R**.

Biofloranate G (**3**) was also obtained as a colorless oil. The molecular formula of C_21_H_36_O_3_ and four degrees of unsaturation were inferred from its HRESIMS at *m*/*z* 319.2637 [M − H_2_O + H]^+^ (calcd. 319.2637). Examination of the ^1^H and ^13^C NMR spectrum revealed that the difference between compounds **2** and **1** was due to the presence of an ester carbonyl group (*δ*_C_/*δ*_H_ 177.6 and 51.6/3.67) at C-15 in **2**, replacing the oxygenated methylene group found in **1**. This was corroborated by the corresponding HMBC correlations (Figure 2). The relative stereochemistry of **3** was also determined through key NOE interactions of H-5/H_3_-20, H-5/H-10 and H-5/H_3_-18, and no NOE was detected between H-5 and H-6. Considering biogenetic information, the absolute configuration of **3** was established. However, the stereochemistry at C-15 remained unresolved.

Biofloranate H (**4**), a colorless oil, was found to have a molecular formula of C_21_H_34_O_3_, as determined by HRESIMS at *m*/*z* 317.2496 [M − H_2_O + H]^+^ (calcd. 317.2481), indicating five degrees of unsaturation. The 1D NMR data of **4** closely resembled those of **3**, except for an additional double bond characterized by chemical shifts *δ*_C_/*δ*_H_ of 142.8/6.76 and 127.5. The ^1^H-^1^H COSY correlation of H_2_-13/H-14, and HMBC correlations of H-14/C-12/C-13/C-15/C-16/C-17 (Figure 2), confirmed the presence of an additional double bond between C-14 and C-15 in **4**. The planar structure of **4** was determined. Additionally, compound **5** was identified to have the same molecular formula as **4**, based on HRESIMS analysis. Comparison of the 1D and 2D NMR spectral data of **5** with **4** revealed identical planar structures. The *E* geometry of the 14,16-double bonds in compounds **4** and **5** was inferred from 1D NOE enhancements observed for the H-14/16-OMe moiety. The relative configurations of the two enantiomers were also determined using 1D NOE spectra. For compound **4**, the NOEs of H_3_-18/H-5, H_3_-18/H-6, H-5/H_3_-20, and H-10/H_3_-20 indicated that these protons are on the same face of the molecule. Consequently, the relative configuration of **4** was determined to be 5*R**, 6*R**, 9*R**, 10*S**, 11*R**. Similarly, for compound **5**, the NOEs of H_3_-18/H-5, H-5/H_3_-20, and H-10/H_3_-20 positioned these protons on the same face, while the absence of an NOE correlation between H-6 and H_3_-18 suggested a different relative configuration of 5*R**, 6*S**, 9*R**, 10*S**, 11*R**.

The absolute configurations of compounds **2**, **3**, and **5** were determined by comparing their experimental CD spectroscopic data with that of compound **1**. As depicted in Figure 4a, the CD spectrum of compounds **1**–**3** and **5** exhibited a broad positive Cotton effect at 200 nm. For compound **4**, the experimental ECD spectrum was compared with the calculated spectrum for the (5*R*,6*S*,9*R*,10*S*,11*R*) configuration (Figure 4b), confirming the absolute configuration of **4**.

Lemnabourside H (**6**) was obtained as an amorphous solid. Its molecular formula was determined to be C_30_H_46_O_8_ by HRESIMS (*m*/*z* 557.3082 [M + Na]^+^ (calcd for C_30_H_46_O_8_, 557.3114), requiring eight degrees of unsaturation. The spectral data of **6** closely resembled those of diterpene glycosides isolated from *Lemnalia bournei* [9,10] (Table 2), except for two acetyl groups substituted on distinct hydroxyl groups of the sugar moiety. The D-glucose was confirmed to be in boat form, as reported in [12]. The position of two acetyl groups at 2′- and 3′-OH groups was established through HMBC correlations from H-2′ to 2′-OAc (*δ*_C_ 170.4), and from H-3′ to 3′-OAc (*δ*_C_ 171.5). Acid hydrolysis of **6,** performed according to a previously reported method [12], yielded D-glucose and diterpene aldehyde. The relative stereochemistry of the diterpene unit of **6** was inferred from 1D NOE experiments (Figure 3) and biogenetic consideration.

Lemnabourside I (**7**) was isolated as a yellow solid. It exhibited an HRESIMS ion peak at *m*/*z* 643.3478 [M + Na]^+^, consistent with a molecular formula of C_34_H_52_O_10_. NMR analysis (Table 2) indicated that **7** is a ring D opened diterpene glycoside, similar to lemnaboursides F and G, with all hydroxyl groups of the glucose acetylated. The ^1^H-^1^H COSY, HSQC, HMBC, and 1D NOE experiments facilitated the complete structural assignment of **7** (Figure 2 and Figure 3). Acid hydrolysis of **7** also yielded D-glucose and a diterpene aldehyde. The coupling constant of the anomeric proton (1H, d, *J* = 8 Hz) suggested that the glucose moiety of **7** was in chair form [12]. The absolute configurations of **6** and **7** were deduced by comparison of the experimental CD spectroscopic data with the known compound **8**, which showed a negative Cotton effect at 197 nm (Figure 4c).

The antibacterial activities of compounds **1**–**8** were assessed against five pathogenic bacteria: *Staphylococcus aureus*, *Bacillus subtilis*, *Pseudomonas aeruginosa*, *Streptococcus pneumoniae*, and *Escherichia coli*. As shown in Table 3, all of these diterpenes and diterpene glycosides displayed antibacterial activities against *Staphylococcus aureus* and *Bacillus subtilis*, with MICs ranging from 4 to 64 µg/mL. None of the compounds exhibited an antibacterial effect against *Streptococcus pneumoniae*. Furthermore, the cytotoxicity of the isolated compounds was evaluated in vitro against A549, HeLa, and HepG2 at 20 μM. The results showed that these compounds were inactive against tested cell lines.

## 3. Materials and Methods

### 3.1. General Chemical Experimental Procedures

A ThermoFisher Evolution 201/220 spectrophotometer (Thermo Scientific, Waltham, MA, USA) was used for UV spectroscopy measurements. Optical rotations were measured using a Jasco P-1010 Polarimeter (JASCO, Tokyo, Japan) with sodium light (589 nm). NMR spectra were recorded on a Bruker AVANCE NEO 600 spectrometer (BrukerBiospin AG, Fällanden, Germany). ^1^H chemical shifts were referenced to the residual CDCl_3_ (7.26 ppm), and ^13^C chemical shifts were referenced to the CDCl_3_ (77.2 ppm) solvent peaks. High-resolution electrospray ionization mass spectra (HRESIMS) were performed on an ultra-high-performance liquid chromatograph (UPLC) and TIMS-QTOF high-resolution mass spectrometry (Waters, Milford, MA, USA). The purification was performed by reversed-phase high-performance liquid chromatography using a Shimadzu LC-20AT system (Shimadzu Corporation, Tokyo, Japan). The solvents used for HPLC were all Fisher HPLC grade. A Cosmosil C_18_-MS-II column (250 mm × 20.0 mm, id, 5 μm, Cosmosil, Nakalai Tesque Co., Ltd., Kyoto, Japan) was used for the preparative HPLC separation. Column chromatography was performed using silica gel (300–400 mesh, Qingdao Ocean Chemical Co. Ltd., Qingdao, China) and C_18_ reversed-phase silica gel (75 µm, Nakalai Tesque Co., Ltd., Kyoto, Japan).

### 3.2. Animal Material

Soft coral *Lemnalia bournei* was sampled off the coast of Xisha Islands, South China Sea, 7 m underwater, and frozen immediately after collection. The specimens (XSSC202103) were deposited at the Li Dak Sum Yip Yio Chin Kenneth Li Marine Biopharmaceutical Research Center, Health Science Center, Ningbo University, China.

### 3.3. Extraction and Isolation

The frozen soft coral *Lemnalia bournei* (dry weight: 198.0 g) was freeze-dried and cut into pieces. Then, it was extracted with CH_2_Cl_2_ and MeOH 5 times. The combined extract was evaporated and concentrated to obtain a crude residue, which was then partitioned between Et_2_O and H_2_O. The Et_2_O solution was concentrated under reduced pressure to give a residue (24.0 g). The residue was eluted with petroleum ether/EtOAc (100:0~1:1, *v*:*v*) on a gradient silica gel column chromatography, and five fractions (Fr.1~Fr.5) were obtained. Fr.1 (693.9 mg) was eluted with MeOH/H_2_O (50:50 to 100:0, *v*/*v*) on reversed-phase column chromatography to obtain four subfractions (Fr.1.1-Fr.1.4). Purification of Fr.1.3 (25.8 mg) by semi-preparative HPLC (MeCN/H_2_O, 46:54, 2 mL/min) gave compounds **1** (3.5 mg) and **2** (3.9 mg). Separation of Fr.2 (647.9 mg) on a reversed-phase column with MeOH/H_2_O (50:50~100:0, *v*/*v*) afforded seven subfractions (Fr.2.1~Fr.2.7). Fr.2.5 (13.5 mg) was purified using semipreparative reversed-phase HPLC (MeCN/H_2_O, 72: 28, 2 mL/min) to afford compounds **3** (2.3 mg) and **4** (2.7 mg). Fr.3 (647.9 mg) was eluted with MeOH/H_2_O (40:60~80:20, *v*/*v*) on an ODS column to give seven subfractions (Fr.3.1~Fr.3.7). Fr.3.5 (29.1 mg) was purified using semi-preparative reversed-phase HPLC (MeCN/H_2_O, 72:28, 2 mL/min) to give compound **5** (3.1 mg). Fr.4 (881.5 mg) was separated on an ODS column with MeOH/H_2_O (58:42~100:0, *v*/*v*) to give six subfractions (Fr.4.1-Fr.4.6). Fr.4.2 (19.3 mg) was purified using HPLC (MeCN/H_2_O, 92:8, 2 mL/min) to obtain compound **6** (6.3 mg). Fr.4.3 (15.7 mg) was purified using semi-preparative HPLC (MeCN/H_2_O, 90:10, 2 mL/min) to give compound **7** (4.1 mg). Fr.5 was separated on reversed-phase MPLC using the mobile phase of MeOH-H_2_O (60:40~80:20, *v*/*v*) to obtain compound **8** (125.0 mg).

Biofloranate E (**1**): colorless oil; {[α${]}_{D}^{25}$ + 23.3 (c 0.10, MeOH)}; UV (MeOH) λ_max_ (log ε) 197 (0.84) nm; ^1^H and ^13^C NMR data, Table 1; HRESIMS *m*/*z* 291.2669 [M − H_2_O + H]^+^ (calcd for C_20_H_35_O, 291.2682).

Biofloranate F (**2**): colorless oil; {[α${]}_{D}^{25}$ + 21.8 (c 0.10, MeOH)}; UV (MeOH) λ_max_ (log ε) 198 (1.56) nm; 229 (0.54) nm; ^1^H and ^13^C NMR data, Table 1; HRESIMS *m*/*z* 287.2371 [M − H_2_O + H]^+^ (calcd for C_20_H_31_O, 287.2375).

Biofloranate G (**3**): colorless oil; {[α${]}_{D}^{25}$ + 26.7 (c 0.10, MeOH)}; UV (MeOH) λ_max_ (log ε) 204(0.89) nm; ^1^H and ^13^C NMR data, Table 1; HRESIMS *m*/*z* 319.2636 [M − H_2_O + H]^+^ (calcd for C_21_H_35_O_2_, 319.2637).

Biofloranate H (**4**): colorless oil; {[α${]}_{D}^{25}$ + 19.3 (c 0.10, MeOH)}; UV (MeOH) λ_max_ (log ε) 209 (0.99) nm; ^1^H and ^13^C NMR data, Table 1; HRESIMS *m*/*z* 317.2496 [M − H_2_O + H]^+^ (calcd for C_21_H_33_O_2_, 317.2481).

Biofloranate I (**5**): colorless oil; {[α${]}_{D}^{25}$ + 19.3 (c 0.10, MeOH)}; UV (MeOH) λ_max_ (log ε) 217 (1.05) nm; ^1^H and ^13^C NMR data, Table 1; HRESIMS *m*/*z* 317.2472 [M − H_2_O]^+^ (calcd for C_21_H_33_O_2_, 317.2463).

Lemnabourside H (**6**): amorphous solid; {[α${]}_{D}^{25}$ + 21.66 (c0.10, MeOH)}; UV (MeOH) λ_max_ (log ε)203(0.97) nm; ^1^H and ^13^C NMR data, Table 2; HRESIMS *m*/*z* 557.3082 [M + Na]^+^ (calcd for C_30_H_46_O_8_Na, 557.3091).

Lemnabourside I (**7**): yellow solid; {[α${]}_{D}^{25}$ + 38.2 (c 0.10, MeOH)}; UV (MeOH) λ_max_ (log ε) 199 (1.44) nm. ^1^H and ^13^C NMRdata, Table 2; HRESIMS *m*/*z* 643.3427 [M + Na]^+^ (calcd for C_34_H_52_O_10_Na, 643.3458).

### 3.4. DFT-ECD Calculation

Theoretical ECD spectra of compounds **1** and **4** were calculated using the Gaussian 16 program package. Conformational analysis and density functional theory (DFT) calculations were used to generate and optimize conformations at the B3LYP/6-311G (2d,p) level of theory, following the method reported in [13].

### 3.5. Antibacterial Assays

All the isolated compounds were tested for antibacterial activities by following the literature [14]. Five bacterial strains of *Staphylococcus aureus* [CMCC (B) 26003], *Bacillus subtilis* [CMCC (B) 63501], *Vibrio harveyi* 1708B04 (accession number: MZ333451), *Streptococcus pneumoniae* [CMCC (B) 31001], and *Escherichia coli* [CMCC (B) 44102] were selected, and penicillin G was chosen as a positive control. Compounds **1**–**8** were dissolved in DMSO and tested at concentrations of 128, 64, 32, 16, 8, 4, and 2 µg/mL. Briefly, the bacteria were grown in MH medium for 24 h at 28 °C with agitation (180 rpm) and then diluted with sterile MH medium to match 0.5 McFarland standard. One hundred microliters of each bacterial supernatant and 100 μL of MH medium with 0.002% 2,3,5-triphenyltetrazolium chloride, together with test or control materials, were incubated. Then, the inhibition data were recorded optically.

### 3.6. Cytotoxic Activity Assays

A549, HeLa, and HepG2 cells were cultured in DMEM (Gibco; Thermo Fisher Scientific, Inc., Waltham, MA, USA) supplemented with 10% FBS (Gibco; Thermo Fisher Scientific, Inc.). A549, HeLa, and HepG2 cells were obtained from the Chinese Academy of Science Shanghai Cell Bank. The above cells were cultured at 37 °C under a humidified 95%:5% (*v*/*v*) mixture of air and CO_2_.

MTT assays were performed as previously described [15]. Briefly, cells were seeded into 96-well plates at a density of 5 × 10^3^ cells/well, incubated for 12 h, and then exposed to different test compounds at different concentrations for 48 h. Next, the cells were stained with 20 μL of MTT solution (5 mg/mL) for 4h. Finally, the medium and MTT solution mixture was removed, and 150 μL of DMSO was added to dissolve formazan crystals. It was then shaken at low speed on a shaker for 10 min. The absorbance of each well was measured at 490 nm using a microplate reader. The cell growth curve was plotted with the time abscissa and the absorbance value as the ordinate.

## 4. Conclusions

In summary, the chemical study of *Lemnalia bournei*, a soft coral collected from the South China Sea, resulted in the identification of five novel biflorane-type diterpenoids, biofloranates E-I, two new bicyclic diterpene glycosides, lemnaboursides H-I, and the known lemnabourside. The structures of these compounds were elucidated using NMR spectroscopy and ECD analysis and corroborated existing literature. These compounds demonstrated antibacterial properties against *Staphylococcus aureus* and *Bacillus subtilis*, with MICs ranging from 4 to 64 µg/mL. The discovery of these compounds contributes to the diversity and complexity of terpenoids isolated from marine soft corals.

## Figures and Tables

**Figure 1 marinedrugs-22-00157-f001:**
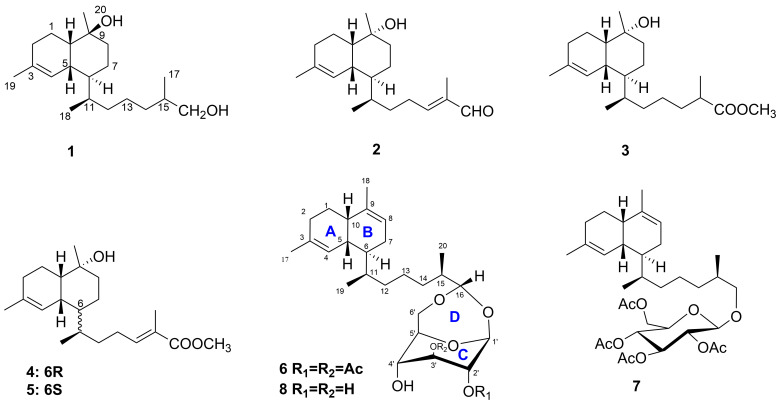
Chemical structures of compounds **1**–**8**.

**Figure 2 marinedrugs-22-00157-f002:**
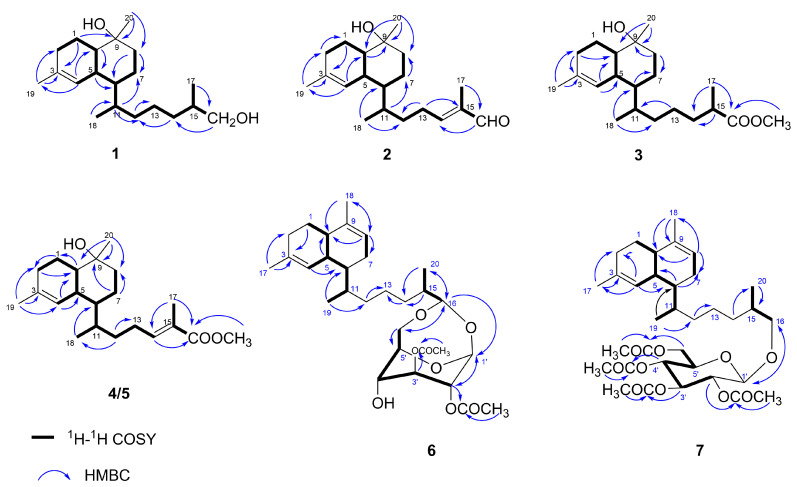
^1^H-^1^H COSY and key HMBC correlations of compounds **1**–**7**.

**Figure 3 marinedrugs-22-00157-f003:**
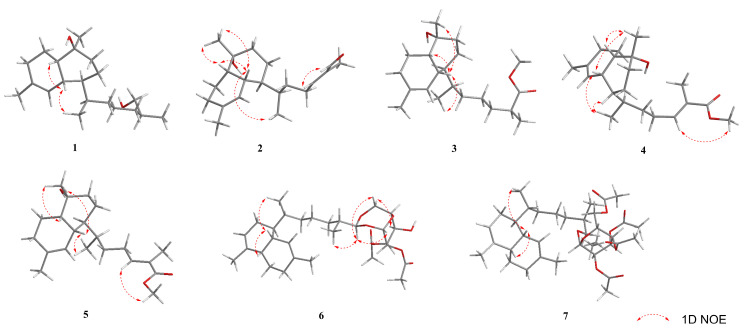
Key NOE correlations of compounds **1**–**7**.

**Figure 4 marinedrugs-22-00157-f004:**
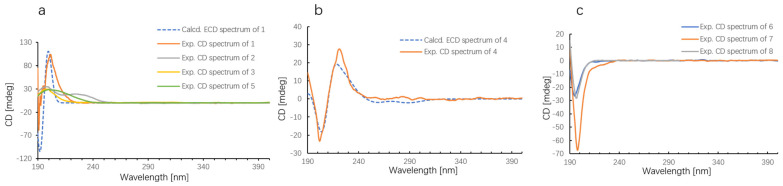
(**a**) Experimental ECD spectra of compounds **1**–**3** and **5** and calculated ECD spectrum of compound **1**. (**b**) Experimental and calculated ECD spectra of compound **4**. (**c**) Experimental ECD spectra of compounds **6**–**8**.

**Table 1 marinedrugs-22-00157-t001:** ^1^H NMR and ^13^CNMR data for compounds **1** to **5** at 600 MHz in CDCl_3_.

No.	1		2		3		4		5	
	*δ*_H_ (*J* in Hz)	^13^C	*δ*_H_ (*J* in Hz)	^13^C	*δ*_H_ (*J* in Hz)	^13^C	*δ*_H_ (*J* in Hz)	^13^C	*δ*_H_ (*J* in Hz)	^13^C
1	1.58, m *	21.1, CH_2_	1.91, m *	18.6, CH_2_	1.89, m *	18.6, CH_2_	1.25, m *	22.8, CH_2_	1.52, m *	18.6, CH_2_
			1.52, m *		1.54, m *		2.01, m *		1.90, m *	
2	1.54, m *	31.4, CH_2_	1.98, m *	31.3, CH_2_	1.99, m *	31.3, CH_2_	1.97, m	31.1, CH_2_	1.98, m *	31.3, CH_2_
	1.99, m *				1.28, m					
3		133.7, C		135.0, C		134.6, C		135.4, C		134.7, C
4	5.54, m	124.8, CH	5.47, dq (5.0, 1.6)	124.1, CH	5.49, m	124.5, CH	5.44, s	122.1, CH	5.47, td (3.4, 1.5)	124.3, CH
5	2.30, m	34.3, CH	2.05, m *	36.5, CH	2.02, m *	36.6, CH	1.76, m *	39.7, CH	2.04, m *	36.5, CH
6	1.31, m *	42.5, CH	1.40, m *	42.4, CH	1.34, m *	42.8, CH	1.14, m *	45.0, CH	1.38, m	42.4, CH
7	1.37, m *	19.8, CH_2_	1.42, m *	21.9, CH_2_	1.09, dd (13.0, 4.2)	21.9, CH_2_	1.53, dt (9.9, 3.5)	22.3, CH_2_	1.12, td (13.0, 4.2)	21.9, CH_2_
	1.29, m *		1.15, m *		1.40, dq (13.1, 4.7)		2.00, m *		1.42, m *	
8	1.53, m *	34.8, CH_2_	1.53, m *	35.3, CH_2_	1.52, m *	35.6, CH_2_	1.80, m *	42.3, CH_2_	1.53, m *	35.4, CH_2_
	1.42, m		1.42, m *				1.43, dt (9.8, 3.8)			
9		72.5, C		72.5, C		72.6, C		72.5, C		72.6, C
10	1.54, m *	46.3, CH	1.60, m	45.7, CH_2_	1.59, m *	45.8, CH	1.22, d (10.4)	50.2, CH	1.61, dt (4.0,1.7)	45.7, CH
11	1.74, dq (6.8, 2.8)	32.0, CH	1.80, dd (10.4, 5.7)	31.6, CH	1.71, dq (6.9, 2.9)	31.7, CH	2.00, m *	31.3, CH	1.77, dq (6.9, 2.7)	31.6, CH
12	1.20, s	36.2, CH_2_	1.54, m *	34.5, CH_2_	1.18, m *	36.0, CH_2_	1.38, q (7.7)	34.6, CH_2_	1.35, m	34.6, CH_2_
			1.41, m *							
13	1.37, m	25.3, CH_2_	2.37, m *	27.5, CH_2_	1.17, m *	25.7, CH_2_	2.17, m *	27.1, CH_2_	2.18, m	27.1, CH_2_
			2.30, m *						2.11, m	
14	1.37, m	33.6, CH	6.46, tq (7.4, 1.4)	155.2, CH	1.53, m *	34.3, CH_2_	6.76, tq (7.5, 1.7)	142.8, CH	6.73, tq (7.5, 1.4)	142.9, CH
	1.08, m *				1.36, m *					
15	1.61, m *	35.9, CH		139.3, C	2.43, d (7.0)	39.6, CH		127.5, C		127.4, C
16	3.50, dd, (10.4, 5.7)	68.6, CH_22_	9.38, s	195.5, CH		177.5, C		168.9, C		168.9, C
	3.41, dd (10.5, 6.5)									
17	0.91, d (6.7)	16.8, CH_3_	1.73, s	9.3, CH_3_	1.13, d (7.0)	17.3, CH_3_	1.83, s	12.5, CH_3_	1.82, s	12.5, CH_3_
18	0.83, d (6.9)	13.6, CH_3_	0.85, d (6.9)	13.4, CH_3_	0.78, d (6.9)	13.4, CH_3_	0.79, d (6.9)	13.4, CH_3_	0.83, d (6.9)	13.4, CH_3_
19	1.65, s	23.8, CH_3_	1.64, s	23.8, CH_3_	1.66, t (1.1)	23.8, CH_3_	1.67, s	24.0, CH	1.64, s	23.7, CH_3_
20	1.20, s	29.5, CH_3_	1.30, s	28.1, CH_3_	1.29, s	28.1, CH_3_	1.10, s	20.9, CH_3_	1.29, s	28.1, CH_3_
21					3.67, s	51.6, CH_3_	3.72, s	51.8, CH_3_	3.72, s	51.8, CH_3_

* Overlapped.

**Table 2 marinedrugs-22-00157-t002:** ^1^H NMR and ^13^CNMR data for compounds **6** and **7** at 600 MHz in CDCl_3_.

No.	6		7	
	*δ*_H_ (*J* in Hz)	^13^C	*δ*_H_ (*J* in Hz)	^13^C
1	1.84, dd *	24.8, CH_2_	1.35, m	24.8, CH_2_
	1.20, m		1.73, m	
2	1.97, m	31.0, CH_2_	1.97, dd (12.2, 3.7)	31.0, CH_2_
	1.92, m		1.93, m	
3		134. 6, C		134.7, C
4	5.48, m	124.0, CH	5.48, d (4.8)	124.0, CH
5	2.03, m	36.5, CH	2.04, d (6.4)	36.5, CH
6	1.49, m *	39.1, CH	1.49, m *	39.2, CH
7	1.84, dd *	24.8, CH_2_	1.84, m *	24.8, CH_2_
	1.20, m		1.77, m	
8	5.45, m	121.6, CH	5.40, d (4.7)	121.6, CH
9		136.8, C		136. 8, C
10	1.92, m	39.7, CH	1.93, dd (12.2, 3.7)	39.7, CH
11	1.77, m	31.9, CH	1.77, q (8.4, 6.1)	32.0, CH
12	1.21, m	36.0, CH_2_	1.18, m	36.1, CH_2_
	1.14, m		1.14, m	
13	1.77, m	25.3, CH_2_	1.77, m	25.1, CH_2_
	1.36, m		1.19, m	
14	1.47, m * 1.10, m	31.9, CH_2_	1.34, m * 1.05, q (9.0)	33.8, CH_2_
15	1.70, m *	38.1, CH	1.69, q (8.4, 6.1)	33.3, CH
	1.14, m			
16	4.59, d (4.7)	102.0, CH	3.65, td (9.2, 5.5)	75.6, CH_2_
			3.29, dd (9.5, 6.2)	
17	1.69, m *	24.1, CH_3_	1.69, q (3.5)	24.1, CH_3_
18	1.68, m *	21.9, CH_3_	1.67, m	21.9, CH_3_
19	0.81, d, (6.7)	13.5, CH_3_	0.80, d (6.8)	13.5, CH_3_
20	0.92, d (6.8)	14.4, CH_3_	0.86, t (7.5)	16.8, CH_3_
1′	4.87, br.s	98.6, CH	4.46, d (8.0)	101.2, CH
2′	4.89, d (5.9)	75.8, CH	4.99, dd (9.6, 8.0)	71.5, CH
3′	5.15, dd (10.9, 5.9)	75.4, CH	5.20, t (9.5)	73.0, CH
4′	4.41, dd (10.9, 7.8)	65.0, CH	5.09, t (9.7)	68.7, CH
5′	3.93, dt (7.8, 1.7)	80.2, CH	3.67, td (9.2, 5.5)	71.9, CH
6′	3.96, dd (12.6, 1.3)	67.7, CH_2_	4.26, dd (12.2, 4.7)	62.1, CH_2_
	3.56, dd (12.6, 2.0)		4.13, dd (12.3, 2.5)	
2′-OAc	2.09, s	170.4, C 21.1, CH_3_	2.02, d (6.4)	169.4, C 20.8, CH_3_
	2.09, s	21.1, CH_3_	2.02, d (6.4)	20.8, CH_3_
3′-OAc 3′-OAc	2.12, s	171.5, C 21.2, CH_3_	2.01, s	170.5, C 20.8, CH_3_
	2.12, s	21.2, CH_3_	2.01, s	20.8, CH_3_
4′-OAc				169.6, C
			2.03, s	20.8, CH_3_
6′-OAc				170.9, C
			2.08, s	20.9, CH_3_

* Overlapped.

**Table 3 marinedrugs-22-00157-t003:** Antimicrobial activity of compounds **1**–**8**.

Compounds	MIC (Against *Staphylococcus* *aureus*, µg/mL)	MIC (Against *Bacillus subtilis*, µg/mL)	MIC (Against *Vibrio harveyi*, µg/mL)	MIC (Against *Streptococcus pneumoniae*, µg/mL)	MIC (Against *Escherichia coli,* µg/mL)
**1**	32	32	64	>128	>128
**2**	32	32	64	>128	128
**3**	64	32	>128	>128	>128
**4**	64	64	>128	>128	>128
**5**	64	32	>128	>128	>128
**6**	16	16	32	>128	8
**7**	32	16	64	>128	32
**8**	8	4	32	>128	8
Penicillin ^a^	<0.5	<0.5	<0.5	8	<0.5

^a^ Positive control.

## Data Availability

The original data presented in the study are included in the article/Appendix A; further inquiries can be directed to the corresponding author.

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
