# Peer review of "New Diterpenes and Diterpene Glycosides with Antibacterial Activity from Soft Coral Lemnalia bournei"

_marinedrugs, 2024, doi:10.3390/md22040157_

Round 1

Reviewer 1 Report

Comments and Suggestions for Authors

The authors are asked to reflect on and answer the following questions.

1.     The grammar, syntax, spelling, format, style, and/or typography need improvement. Therefore, it is advised that the authors check the material to fix these mistakes. Despite the fact that soft coral names should be italicized, keep in mind that not all references do.

2.     The reference mentioned on page 1, line 36, specifically discusses cemberanoid diterpenes, not biflorane diterpenes.

3.     I recommend that the author modify the manuscript title to "New Diterpenes and Diterpene Glycosides with Antibacterial Activity from Soft Coral Lemnalia bournei" instead of the present one.

4.      Refer to page 2, line 53. It is inappropriate to use the word "discovery," so you have to choose another phrase. Additionally, there are five new compounds, not four.

5.     Please verify skeleton numbering of compound 1, particularly carbons 16 and 17, as they may be reversed. This information can be found on Page 2, Figure 1.

6.     On page 2, line 68, there are HMBC correlations observed not only with quaternary carbons but also with other carbons that have protons attached to them.

7.     Can you confirm the authenticity of compounds 3 and 4?

8.     Why is the anomeric proton in compound 6 in Table 2's splitting pattern broad singlet?

9.     In Figure 2 on Page 4, there is an excessive number of arrows that cause confusion in interpreting the HMBC correlations. It is recommended to include just the significant arrows and exclude the two-bond correlations. Additionally, please review and correct any arrows that have two heads.

10. Are you certain that the freeze-dried sample weight on page 8, line 193 is only 98.0 grams?

11.Page 9, line 255: I prefer to use a new metabolite instead of the discovery in this instance.

12.Please reduce the intensity of the cross peaks in the 2DNMR spectra in the Supporting data section.

with regards and best wishes

Comments on the Quality of English Language

The proficiency of the English language utilized in the manuscript is satisfactory.

Author Response

Dear Reviewer,

Thank you very much for the helpful comments on our manuscript (marinedrugs-2937748). Now, we have revised the manuscript according to the comments and highlighted it in yellow color. The point-by-point responses to all the comments are listed as attached.

Reviewer 2 Report

Comments and Suggestions for Authors

The paper submitted for review contains extensive material described on several pages and a huge set of supplementary materials in the form of one- and two-dimensional NMR spectra, the results of the Nuclear Overhauser Effect (NOE) analysis and other.

It takes more time to review all attachments and validate them. I didn't find any distortions in the randomly selected examples, so I assume that the rest is OK as well.

The Authors rely heavily on their previous published results, so I have no way of verifying whether the sugar-containing systems described are actually derivatives of D-glucose or not.

I am puzzled by the omission of a non-destructive analysis of IR spectra, which would easily indicate the presence of hydroxyl, carbonyl, ether and double bond groups. If it turns out that the IR does not confirm the proposed structure, it would be a problem that should be taken into account at any time.

The Authors systematically examined the biocidal activity of the isolated compounds, proving that it is small compared to the standard in the form of penicillin, which does not inspire optimism for possible use in medicine. It is a pity that the Authors did not use drug-resistant strains, then the detected activity would have been much more valuable.

In the abstract, the Authors write about cytotoxicity tests, including those against some cancer cells, but in the text of the paper I did not find any information on this subject. The study of antibacterial activity was based on a colorimetric biochemical assay using triphenyltetrazolium chloride, while more reliable results are obtained by the culture method. I'm not saying that the results are bad, but I'm not sure.

Finally, about the inconsistencies of the isolated compounds: In lines 58, 88, 102, 112 it is stated that the compounds F(1-4) are colorless oils, while in lines 215, 218, 221, 224 the same compounds are described as white solids. So how is it?

At this point, another remark comes to mind: if these are solids (I assume so), then it would be necessary to obtain single crystals and perform X-ray structural tests, then there would be no doubts about the obtained structure obtained as a result of the deduced structure.

To sum up, the work is interesting, contains a lot of analysis and should be published, and my comments only indicated the direction of its improvement.

Author Response

(The authors gave the same response as above.)
